# PIK Your Poison: The Effects of Combining PI3K and CDK Inhibitors against Metastatic Cutaneous Squamous Cell Carcinoma In Vitro

**DOI:** 10.3390/cancers16020370

**Published:** 2024-01-15

**Authors:** Jay R. Perry, Benjamin Genenger, Amarinder Singh Thind, Bruce Ashford, Marie Ranson

**Affiliations:** 1School of Chemistry and Molecular Bioscience, Molecular Horizon, Faculty of Science, Medicine and Health, University of Wollongong, Wollongong, NSW 2522, Australia; bg038@uowmail.edu.au (B.G.); athind@uow.edu.au (A.S.T.); 2Illawarra Shoalhaven Local Health District, Wollongong, NSW 2500, Australia; bgashford@gmail.com; 3Graduate School of Medicine, University of Wollongong, Wollongong, NSW 2522, Australia

**Keywords:** cutaneous squamous cell carcinoma, cSCC, metastasis, targeted therapy, combination therapy, phosphoinositid-3-kinase, PI3K, PIK-75, cyclin-dependent kinase, CDK1/2/5/9, dinaciclib

## Abstract

**Simple Summary:**

Cutaneous squamous cell carcinoma (cSCC) is a very common skin cancer with poor prognosis for patients with advanced disease. PI3K/AKT/mTOR and cell cycle signalling pathways are often dysregulated in mcSCC. A combination drug approach targeting both pathways concurrently has been theorised to overcome the underwhelming clinical performance of targeted inhibitors individually. This study investigates the potential of PI3K inhibitors (PI3Ki) and cell-cycle inhibitors (CDKi) as single agents and in combination against patient-derived mcSCC cell lines. Whilst PI3Ki and CDKi as single agents potently induced cancer cell death, PI3Ki synergistically enhanced the potential of dinaciclib to induce cell death in one mcSCC cell line, but not another. Interestingly, this pattern was reversed in more complex cell culture models. PI3Ki and CDKi effectively stopped the cell cycle and induced programmed cell death both individually and in combination. These findings suggest that personalised medicine approaches targeting PI3K and CDK pathways in combination may yield some benefit, although further investigation is required to address discrepancies between simple and more complex culture models.

**Abstract:**

Cutaneous squamous cell carcinoma (cSCC) is a very common skin malignancy with poor prognosis for patients with locally advanced or metastatic cSCC (mcSCC). PI3K/AKT/mTOR and cell cycle signalling pathways are often dysregulated in mcSCC. A combination drug approach has been theorised to overcome the underwhelming clinical performance of targeted inhibitors as single agents. This study investigates the potential of targeted inhibition of the p110α−subunit of PI3K with PIK-75 or BGT226 (P13Ki), and of CDK1/2/5/9 with dinaciclib (CDKi) as single agents and in combination. The patient−derived mcSCC cell lines, UW-CSCC1 and UW-CSCC2, were used to assess cell viability, migration, cell signalling, cell cycle distribution, and apoptosis. PIK-75, BGT226, and dinaciclib exhibited strong cytotoxic potency as single agents. Notably, the non-malignant HaCaT cell line was unaffected. In 2D cultures, PIK-75 synergistically enhanced the cytotoxic effects of dinaciclib in UW-CSCC2, but not UW-CSCC1. Interestingly, this pattern was reversed in 3D spheroid models. Despite the combination of PIK-75 and dinaciclib resulting in an increase in cell cycle arrest and apoptosis, and reduced cell motility, these differences were largely negligible compared to their single-agent counterpart. The differential responses between the cell lines correlated with driver gene mutation profiles. These findings suggest that personalised medicine approaches targeting PI3K and CDK pathways in combination may yield some benefit for mcSCC, and that more complex 3D models should be considered for drug responsiveness studies in this disease.

## 1. Introduction

Cutaneous squamous cell carcinoma (cSCC) accounts for approximately 30% of non−melanoma skin cancers. Primary cSCC is generally treated by minor surgical excision or the application of a topical ointment. Age−standardised incidence in mainland Australia has been reported to be as great as 856/10^5^/year, with age−specific rates for men over 60 years old at 2875/10^5^/year, eclipsing melanoma’s age−standardised rate of 69/10^5^/year [1]. Despite a seemingly low metastasis rate (up to 5%), the high incidence of primary cases amounts to a significant burden of metastatic disease. Loco−regional and distant metastases are associated with a worse prognosis due to limited effective therapeutic options compared with other cancers [2]. The immune checkpoint inhibitor cemilipimab has made remarkable strides in the treatment of multiple cancers, with up to 47% response rate in locally advanced and metastatic cSCC [3,4,5,6,7]. However, immunotherapy is contraindicated in immunocompromised patients, and approximately 50% of non−immunosuppressed patients do not benefit [8]. Due to the limitations of immunotherapy in cSCC, other therapeutic options are needed. EGFR inhibitors have shown promise, but despite EGFR overexpression in cSCC, responses can vary regardless of mutation status [8,9]. Further, resistance to EGFR−targeted therapies has been documented—a result of sustained signalling downstream of the EGFR axis via, for example, constitutively active *RAS* or *PIK3CA* mutants [8,10]. Consequently, the limited availability of therapies and shortcomings of current therapies for mcSCC warrants the development of novel targeted therapies.

The development of novel targeted therapies hinges on the mutational landscape of a tumour. Genetic alterations to components of the PI3K/AKT/mTOR signalling network are common across many cancers, including in cSCC [11,12,13]. Notably *PIK3CA*, encoding for the p110α catalytic subunit of phosphatidylinositol 3−kinase (PI3K), has been shown to be mutated in 6–21% of aggressive cSCC [14,15,16]. Isoforms belonging to the class I subgroup of PI3K have key regulatory and catalytic roles, activating AKT, a serine/threonine kinase. In a 140-patient cohort, cSCC tumours displayed significantly increased activation of AKT (89% increase) and expression of EMT markers compared to normal skin [17,18]. AKT pathway activation in mcSCC can be independent of any one somatic mutation and occur because of miRNA or upstream regulators [19,20]. Upon activation, AKT triggers a cascade of tumorigenic signalling, influencing cell survival, cell cycle progression, cancer cell metabolism, and angiogenesis. PI3K/AKT/mTOR−targeting agents have therefore presented an attractive therapeutic for cSCC, but have been met with challenges [21]. Chiefly, varying sensitivity towards cSCC (and other solid tumours) has been shown depending on the isoform−specificity of the PI3K inhibitor (PI3Ki) [2]. Patients lacking mutations in *PIK3CA* are far less likely to respond positively to PI3Ki in general, but particularly those specific to targeting p110α [22]. Additionally, patients treated with a PI3Ki specific to their mutation may demonstrate no response due to other mechanisms of resistance. A first generation of pan−PI3Ki has been investigated in clinical trials and had an acceptable safety profile but at best only modest activity as a monotherapy [23]. However, the emergence of isoform specific inhibitors allowed second−generation PI3Ki to be tailored more specifically towards the specific genomic landscape of the tumour, e.g., p110α inhibitors in cSCC [19,20,24]. PIK-75, a p110α inhibitor, has shown efficacy in both in vitro and in vivo cancer models [25], although its tolerability in clinical practice may preclude its use at effective doses as a single agent [26]. Advantageously, PIK-75 is also a potent inhibitor of DNA−PK, which has implications in DNA damage repair (DDR), telomere maintenance, and basal transcription [25,27,28].

Dysregulation of the cell cycle is also a hallmark of neoplastic cells [29], and therefore, a desirable drug target [30]. The transition through the cell cycle checkpoints between the different growth phases is regulated by cyclin−dependent kinases (CDKs) and alternating levels of various cyclins. First−generation pan−CDK inhibitors (CDKi) including flavopiridol (CDK1/2/4/6/7/9) performed well in vitro but showed low tolerability in clinical trials [29]. Use as a monotherapy has largely been discontinued in favour of combination strategies [31,32,33]. Second−generation selective CDKis have proven more efficacious and tolerable, including the USA Food and Drug Administration−approved palbociclib (CDK4/6) in combination with anti−oestrogen therapies for breast cancer [34]. Dinaciclib is a pan−CDKi with nanomolar potency towards CDK1, CDK2, CDK5, and CDK9 [35] and potent activity against a range of human cancer cell lines with promising results in preclinical safety and pharmacokinetic studies [35]. Phase I–III clinical trials involving dinaciclib report favourable tolerability and safety but mixed therapeutic success [36,37,38].

We have previously shown a potent response to PIK-75 (PI3Ki) as well as dinaciclib (CDKi) as single agents against cell lines derived from nodal metastases of cSCC from the head and neck region [39]. However, the individual shortcomings of each drug, especially the shared susceptibility to resistance, highlights the benefits of a combinatorial approach. Synergistic effects of CDK and PI3K/AKT inhibition have been observed in pre−clinical studies on some cancers [40,41], although none have investigated this with mcSCC. Hence, the aim of the current study is to evaluate the benefit of combining the PI3Ki, PIK-75, with the CDKi, dinaciclib, in our mcSCC cell lines, as representative models of mcSCC.

## 2. Materials and Methods

### 2.1. General Cell Culture and Maintenance

This study utilises two metastatic cSCC cell lines, UW-CSCC1 and UW-CSCC2, established from lymph node cSCC metastases in two patients. The genotypic and phenotypic characterisation of these cell lines is described in detail Perry, Ashford, Thind, Gauthier, Minaei, Major, Iyer, Gupta, Clark and Ranson [39]. For preliminary evaluation of drug cytotoxicity on a non−tumorigenic cell line, the spontaneously immortalised human keratinocyte cell line HaCaT [42] was utilised. Both UW-CSCC1 and HaCaT cells were routinely cultured in Dulbecco’s Modified Eagle Medium (DMEM, ThermoFisher Scientific, Waltham, MA, USA) supplemented with 10% foetal calf serum (FCS, Cellsera, Australia), glucose (4500 mg/mL), and penicillin/streptomycin (50 U/mL, Gibco, Life Technologies Cooperation, Carlsbad, CA, USA), with the addition of 20 ng/mL human epidermal growth factor (hEGF) to the HaCaT growth medium. UW-CSCC2 were cultured in Advanced DMEM/F12 (ThermoFisher Scientific, USA) supplemented with 20 ng/mL hEGF (Gibco, Life Technologies Cooperation, USA), 2% FCS, 1% L-glutamine (Gibco, Life Technologies Cooperation, USA), and penicillin/streptomycin (50 U/mL). Unless otherwise stated, all cells were cultured at 37 °C under hypoxic (3% O_2_) conditions. All experiments were conducted in complete medium and low−oxygen conditions to best mimic tumour physiological conditions. Normoxic (~21% O_2_) acclimated populations of cells were necessary for experiments involving live−cell imaging with the IncuCyte^®^ Zoom kinetic imaging system (Essen BioScience, Ann Arbor, MI, USA), due to accessibility limitations. This was achieved by incubating under standard atmospheric oxygen levels for a minimum of 72 h prior to experimentation. Cultures were regularly screened for mycoplasma contamination using a MycoAlert™ mycoplasma test kit (Lonza Group, Basel, Switzerland).

### 2.2. Nucleic Acid Extraction, Sequencing, and Analysis

Cell line DNA and RNA were extracted as per previously described [39]. All nucleic acid samples were quantified using the NanoDrop spectrophotometer (ND1000, ThermoFisher Scientific, USA) and met the purity requirements for downstream applications (A260/280 between 1.7 and 2.3). RNA was confirmed via agarose gel electrophoresis to be free of DNA contamination and integrity determined via a Qubit™ RNA IQ assay (ThermoFisher Scientific, USA) using the Qubit™ 3.0 fluorometer (ThermoFisher Scientific, USA). Samples with an RNA IQ score < 5 were not considered for downstream analysis.

Whole transcriptome RNA sequencing was conducted by AGRF (Australia) using strand−specific RNA sequencing library preparation (50 million reads) and samples run on the HS4000 Illumina system (Illumina, San Diego, CA, USA). Whole genome sequencing was performed on Illumina HiSeqX instruments (Illumina, USA) by Genome.One Pty Ltd., at the Kinghorn Centre for Clinical Genomics, Garvan Institute of Medical Research (Darlinghurst, NSW, Australia) and analysed as described in Perry, Ashford, Thind, Gauthier, Minaei, Major, Iyer, Gupta, Clark and Ranson [39].

RNA sequencing data were pre−processed using a custom pipeline designed to remove sequencing artefacts and obtain high−quality reads for downstream analysis. Firstly, adapter sequences were trimmed from the reads using cutadapt (v2.4) with a minimum Phred score of 20. Additionally, consecutive bases with a quality score below 20 from the 3′ end of the reads were removed to eliminate low−quality bases that may affect downstream analyses. Next, reads were aligned to the human genome (GRCh38) using the splice−aware aligner STAR (v2.7.2). The option --quantMode TranscriptomeSAM was used to output alignments in transcriptomic coordinates, allowing for gene expression quantification at the transcript level. The transcript database GENCODE Genes (v31) was used as a reference annotation. Reads mapping to multiple locations were allowed, and gene−level expression counts and transcripts per million (TPM) were estimated using RSEM. Heatmaps of gene expression data were visualised using pheatmap [43].

### 2.3. Identification of Recurrently Mutated and Relevant Genes

In this study, we utilised genomic data from a clinical cohort of 25 metastatic cSCC that we have previously published [20]. Genes mutated in at least 30% of samples in this cohort were considered recurrently mutated, and those strongly associated with PI3K/AKT/mTOR, cell cycle, and apoptosis signalling were selected as genes of interest. UW-CSCC1 and UW-CSCC2 molecular data were interrogated for alterations in these genes of interest, including variant type, copy number and gene expression changes. Combined annotation−dependent depletion (CADD) scores were used as a measure of biological impact for each variant, with scores below 20 believed to indicate minimal impact.

### 2.4. Drug Sourcing and Storage

PIK-75, BGT226, and dinaciclib were obtained as powder (Selleckchem, Houston, TX, USA, Cat no. S1205; S2749; S2768) and stock solutions prepared using DMSO and stored at −80 °C. Working stocks were prepared serially, with a final DMSO concentration no greater than 0.2%.

### 2.5. Two−Dimensional Single−Agent Drug Viability Assay

UW-CSCC1, UW-CSCC2, and HaCaT cells were treated with serial dilutions of either PI3Ki or CDKi, in triplicate at a minimum. Metabolic activity of cells was determined after 72 h at 37 °C using the MTS−based CellTitre 96^®^ Aqueous One Solution Cell Proliferation Assay (Promega, Madison, WI, USA, Cat no. G3581) according to manufacturer’s instructions and as previously described [44]. The raw data of treated cells were normalized against vehicle controls with background absorbance values subtracted. Half−maximal inhibitory concentration (IC_50_) values were derived with GraphPad Prism V5 (GraphPad Software, Boston, MA, USA) using a logarithmic sigmoidal dose–response curve fit with a variable slope.

### 2.6. Two−Dimensional Drug Combination Synergy Assay

A checkerboard assay was performed in triplicate to evaluate the benefits of a combination of PIK-75 with dinaciclib (henceforth PIK:DIN) and BGT226 with dinaciclib (henceforth BGT/DIN) against UW-CSCC1 and UW-CSCC2. The concentration range of drugs was based around the IC_50_ values determined from the single−agent screen. Cell viability was determined using the CellTitre 96^®^ Aqueous One Solution Cell Proliferation Assay as per the single−agent assays. Combinations of PIK-75 with other chemotherapeutics including 5−fluorouracil, KX2−391 (SRC inhibitor), and gefitinib (EGFR inhibitor) were similarly assessed, and are available in Appendix A).

A fixed−ratio drug combination screen was subsequently used (*n* = 9) to validate patterns of sensitivity found in the checkerboard assay. In this assay, drug effects were evaluated both as single agents and in combination at a constant ratio based on their approximate IC_50_ values as determined from single−agent assays described above. Concentrations of 0.25×, 0.5×, 1×, 2×, and 4× the IC_50_ of each drug were used, as necessary, for the Chou–Talalay median−effect equation [45].

### 2.7. Drug Synergy Analysis

Raw data from the checkerboard assays were normalised as a percentage of the no drug control and used as input with the drug combination analysis software SynergyFinder 3.0 (https://synergyfinder.fimm.fi (accessed on 20 October 2023)). The program was run, and the BLISS model output was examined for evidence of synergy.

Fixed−ratio data were normalised to the respective treatment’s vehicle control and imported into the software CalcuSyn 2.0 (Biosoft, San Francisco, CA, USA). Combination index (CI) and dose−reduction index (DRI) were calculated to determine synergism/antagonism. A CI < 1 indicates synergism, a CI > 1 indicates antagonism, and a CI = 1 indicates an additive effect.

### 2.8. Three−Dimensional Drug Viability Assay

To generate tumour spheroids, UW-CSCC1 and UW-CSCC2 were seeded at 4000 cells per well (*n* = 10) in round−bottomed, ultra−low attachment 96−well plates (Costar^®^ Corning Incorporated, Corning, NY, USA) in their relevant media. The cells were centrifuged at 209× *g* for 3 min to facilitate their aggregation and incubated in a normoxic incubator (~21% O_2_, 37 °C). After 4 days of culture, the spheroids were treated with vehicle control (DMSO), dinaciclib, PIK-75, BGT226, PIK:DIN, or BGT/DIN at concentrations 4 times greater than those determined to be synergistic in 2D: PIK-75: 250 nM; dinaciclib: 64 nM (UW-CSCC1) and 32 nM (UW-CSCC2). In the absence of data regarding the most synergistic concentration of BGT226, a 4−fold increase in the IC_50_ (1000 nM) was used for this treatment.

The spheroids were monitored and imaged every 2 h using the 4× objective of the IncuCyte^®^ Zoom kinetic imaging system (Essen BioScience, USA). Spheroid integrity was assessed using the Incucyte^®^ Cytotox Green assay according to the manufacturer’s instructions. An increase in mean green fluorescence intensity corresponds with a decrease in cell viability.

### 2.9. AnnexinV−FITC Apoptosis Assay

UW-CSCC1 and UW-CSCC2 were seeded into 25 cm^2^ tissue culture flasks at a density of 200,000 cells per flask and allowed to settle overnight. Cells were then treated with either the vehicle control (DMSO), PIK-75 (both cell lines: 62.5 nM), dinaciclib (UW-CSCC1: 16 nM; UW-CSCC2; 8 nM), or in combination for 24 h and then harvested using trypsin−EDTA. Cells were resuspended in ice−cold binding buffer [10 mM HEPES, 140 mM NaCl, 2.5 mM CaCl_2_; pH = 7.4], and treated with the eBioscience™ Annexin V Apoptosis Detection Kit (Invitrogen, Carlsbad, CA, USA) according to the manufacturers protocol. Cells were analysed by flow cytometry at emissions 561 nm (PI) and 488 nm (FITC) using a BD LSRFortessa™ X-20 flow cytometer (BD Biosciences, Franklin Lakes, NJ, USA) collecting 10,000 events per sample. Live cells were gated, and quadrants were applied to the scatter plots. The proportion of events in either early or late apoptosis, as well as cells undergoing necrosis, were interpreted using GraphPad Prism V5 (GraphPad Software, USA). A one−way ANOVA with Tukey’s multiple comparisons test was applied to quadruplicate observations to determine any significant differences between treatments.

### 2.10. Cell Migration Assay

The effect of treatment on cell migration was assessed in a scratch−wound assay using the IncuCyte^®^ Zoom Kinetic Imaging System (Essen BioScience, USA). UW-CSCC1 and UW-CSCC2 were seeded (*n* = 10) onto collagen 1−coated 96−well ImageLock plates (Essen BioScience, USA) and grown to near confluency in their respective growth media. To prevent proliferation and isolate migratory effects, cells were serum−starved prior to investigation of their motility. This was achieved by replacing media in the wells with a 1% FCS analogue of the relevant growth factor−free culture media. After 24 h incubation in this low serum containing media, the cells were scratched according to manufacturer’s instructions using the 96−pin Woundmaker™ (Essen BioScience, USA). The cells were subsequently washed with serum−free media, then incubated in low serum media containing at 37 °C, 5% CO_2_, and imaged over 48 h at ×10 objective to track cell motility and wound width. IncuCyte™ ZOOM software (Essent BioScience, USA) was used to interpret wound width reduction over time. Output was analysed using GraphPad Prism (GraphPad Software, USA), applying a one−way ANOVA with Tukey’s multiple−comparison post−test.

### 2.11. Cell Cycle Analysis

Flow cytometry was used to determine the influence of treatment on cell cycle distribution. Cells were treated in triplicate with either the vehicle control (DMSO), PIK-75, dinaciclib, or in combination for 48 h and then harvested using trypsin−EDTA. Cells were resuspended in ice−cold PBS at a density of 5 million cells/mL and fixed via the addition of ice−cold 70% ethanol. After a 1 h incubation at −20 °C, cells were pelleted, washed twice with PBS, and resuspended in a propidium iodide (PI) staining mix (40 µg/mL PI, 100 µg/mL RNase A, PBS; pH 7.4). After incubation at 37 °C for 1 h, cells were analysed by flow cytometry at 695 nm using a BD LSRFortessa™ X-20 flow cytometer (BD Biosciences, USA) collecting 10,000 events per sample. DNA content analysis was used to extrapolate the proportion of cells in G0/G1, S, and G2/M phases of cell cycle, calculated based on DNA distribution histograms using FlowJo software (V7.1 Tree Star Inc., Ashland, OR, USA).

### 2.12. Cell Lysate Preperation of Western Blot Analysis

Near−confluent UW-CSCC1 and UW-CSCC2 were treated with DMSO, PIK-75 (62.5 nM), and dinaciclib (16 nM or 8 nM, respectively) as single agents or in combination for 1 and 3 h. Whole−cell lysates were prepared, and Western blot was performed as per the use of antibodies listed in Appendix A. Densitometric analysis of captured blots was performed using ImageJ (V1.53) [46] and results were visualized using GraphPad Prism V5 (GraphPad Software, USA), normalised to GAPDH controls.

## 3. Results

### 3.1. Multi−Omic Analysis of mcSCC Cell Lines

Molecular features of PI3K/AKT/mTOR, cell cycle, and apoptosis signalling pathway elements within mcSCC cell lines UW-CSCC1 and UW-CSCC2 were examined to identify relevant alterations. For this purpose, we identified recurrently mutated genes (≥30% of cases) regulating these pathways in our clinical cohort of metastatic cSCC (*n* = 25, Appendix A). Both cell lines recapitulate many of these recurrent coding mutations, as shown in in Table 1. Interestingly, neither cell line had any of the recurrently mutated apoptosis−related genes, *PRF1* and *CASP8*, nor the pertinent gene *PIK3CA*. UW-CSCC1 and UW-CSCC2 present distinct molecular profiles, each with unique alterations among the cell cycle and PI3K/AKT/mTOR pathway genes. CADD scores were used to assess the potential functional impact of these genetic variants. *TP53*, *AKT3*, and *TGFBR1* (stop-gained) variants in the UW-CSCC1 cell line exhibited notably high CADD scores (Table 1). Despite this, UW-CSCC1 *TGFBR1* mRNA expression was not affected, presenting double the TPM than that of UW-CSCC2, which lacked any *TGFBR1* mutation (Table 1). Additionally, UW-CSCC2 displayed high CADD scores for the *NOTCH1* (stop-gained) and *NOTCH2* (stop-gained) variants (Table 1). The effect of the stop-gained *NOTCH2* mutation is reflected in both the CADD score and gene expression analyses, with a TPM count of 67 and 11 for UW-CSCC1 and UW-CSCC2, respectively. This was similarly found for *NOTCH1*. *CDKN2A* demonstrated a 3.6−fold greater TPM in UW-CSCC1 than UW-CSCC2, aligning with the stop-gained mutation in the latter. Gene expression of *TP53* in UW-CSCC2 (153 TPM) than UW-CSCC1 (109 TPM) were not particularly disparate despite the UW-CSCC2 stop-gained mutation. High−impact missense variants in UW-CSCC1 and UW-CSCC2 *TP53* were also apparent. Stop−gained mutation in *WEE1* in UW-CSCC2 was also noted suggesting additional dysregulation of the G2/M checkpoint and a DNA damage response (DDR) in this cell line.

Genes relating to apoptosis, cell cycle regulation and the PI3K/AKT/mTOR pathway has generally high (2 to 7) copy number (Appendix A). However, there were some small copy number differences between the cell lines among targets of dinaciclib. Notably, the CDK9 copy number was higher in UW-CSCC2 (5.83) compared to UW-CSCC1 (4.06). Conversely, CDK5 amplification was observed in UW-CSCC1 but not UW-CSCC2 (2.93 and 2.00, respectively). The other CDKs targeted by dinaciclib, CDK1 and CDK2, had three gene copies present in both cell lines. The catalytic subunit of PI3K and drug target of PIK-75, *PIK3CA*, had slightly higher copy numbers in UW-CSCC1 (4.60) compared to UW-CSCC2 (3.39), associated with changes in TPM (Appendix A).

Protein expression of key PI3K/AKT/mTOR regulators was also assessed using bead−based analysis, finding incongruence between mRNA level observations (Appendix A). Specifically, phosphorylated mTOR relative to total mTOR protein expression was four−fold higher in UW-CSCC1 than in UW-CSCC2 (Appendix A), despite no notable changes at the mRNA level (Appendix A).

### 3.2. Two−Dimensional Cell Viability Assays and Drug Synergy Studies

In vitro sensitivity of UW-CSCC1 and UW-CSCC2 cells towards the PI3K inhibitors, PIK-75 and BGT226, was determined along with the CDK inhibitor, dinaciclib. In a 2D culture, both cSCC cell lines possessed low nM sensitivity towards dinaciclib (Table 2). UW-CSCC1 were ~four−fold less sensitive to PIK-75 than UW-CSCC2 but showed similar sensitivity towards BGT226 (Table 2). Non−cancerous HaCaT cells produced no observable cytotoxic effect in response to PIK-75 and dinaciclib at the concentrations tested.

A 2D checkerboard assay was used for an initial evaluation of synergism with the PIK:DIN and BGT/DIN combinations. BLISS modelling revealed mild synergistic behaviour at specific concentrations of the PIK:DIN mixtures, with antagonism apparent at extremely high and extremely low concentrations (Figure 1a). Synergism was most evident with 62−125 nM PIK-75/8−16 nM dinaciclib and 62 nM PIK-75/8 nM dinaciclib for UW-CSCC1 and UW-CSCC2, respectively (Figure 1a).

In line with Bliss modelling results, the Fa−CI plots derived from fixed−ratio 2D synergy studies showed additive interactions for PIK:DIN in UW-CSCC1 at lower concentrations but strong antagonism at high Fa (CI >> 1) (Figure 1b; Appendix A). The Fa−CI plots for UW-CSCC2 showed strong synergism at all concentrations of PIK-75 and dinaciclib as most data points fell below CI = 1 (average CI value = 0.5) (Figure 1b; Appendix A). These effects translate to a dose−reduction index (DRI) of 1.5 (PIK-75) and 1.6 (dinaciclib) at Fa 0.5 for UW-CSCC1, and 1.2 (PIK-75) and 10.7 (dinaciclib) for UW-CSCC2 (Figure 1c). The mild DRI of PIK-75 with UW-CSCC1 likely reflects the higher resistance of these cells to this drug alone.

In contrast, the pan−PI3K/mTOR inhibitor BGT226 in combination with dinaciclib produced a mildly antagonistic drug combination profile for both UW-CSCC1 and UW-CSCC2 at the concentrations tested (Figure 2a–c; Appendix A). As a consequence, BGT226 was not considered in further 2D analyses.

### 3.3. Combination PI3K−CDKi Effect on 2D Cell Apoptosis, Cell Cycle Distribution, and Motility

The most synergistic drug combination concentrations identified from the checkerboard assay (Figure 1a) were used to determine the effect upon apoptosis, cell cycle distribution, and migration in comparison to single agents for cSCC cell lines. Despite promising results in a checkerboard assay, these combined concentrations of PIK-75 and dinaciclib failed to produce any significant change in apoptosis (both early apoptosis, late apoptosis, and necrosis) over and above that obtained with the single agents (Figure 3). Whilst the combination did result in a 200% and 20% increase in apoptotic events relative to the control for UW-CSCC1 and UW-CSCC2, respectively, these changes were deemed insignificant (*p* > 0.05). The topoisomerase II inhibitor, etoposide, included as a positive control of apoptosis, expectantly producing a significantly greater proportion of cells undergoing apoptosis than the control (*p* < 0.01) for UW-CSCC1. Interestingly, UW-CSCC2 was not significantly affected by etoposide, although the proportion of necrosis did increase with this agent as well as with the combination compared to controls. The effects of PIK-75 appear to be the driving factor in the response to the drug mixture. However, as the combination with dinaciclib did produce a minor increase in apoptosis, dose optimisation or additional time points may yield more positive results.

Accumulation of propidium iodide was used to determine cell cycle distribution in response to treatment. As single agents, PIK-75 and dinaciclib increased UW-CSCC1 in the S−/G2−phase by 1.63−fold and 1.52−fold, respectively, compared to the controls (Figure 4a,b), mostly due to the arrest in the S−phase. The PIK:DIN mixture increased UW-CSCC1 S−/G2−phase accumulation 1.48−fold compared to the control, mostly due to G2 accumulation (Figure 4a,b). Dinaciclib alone had minimal effects on any cell cycle phase in UW-CSCC2, whereas the most prominent effect of PIK-75 on UW-CSCC2 was G2 arrest (2.1−fold increase compared to control) (Figure 4c,d). The effects of the PIK:DIN mixture were similar to that of PIK-75 alone (Figure 4c,d).

Overall, across both cell lines, there was no significant additive effect on S−/G2−phase accumulation with the PIK:DIN mixture over PIK-75 alone, suggesting the PIK-75−mediated effect dominates the cell cycle stalling within the drug combination.

To assess cell motility in response to the treatments, a scratch wound assay was implemented (Figure 5). Dinaciclib caused almost complete inhibition of UW-CSCC1 cell motility compared to untreated controls with a small but significant effect on UW-CSCC2. However, this potent effect on UW-CSCC1 motility was lost by treatment with PIK:DIN and not enhanced on UW-CSCC2 cells over and above that of dinaciclib alone. PIK-75 alone also caused a moderate decrease in wound confluency compared to the control for both cell lines, with no significant added benefit apparent in the mixture (Figure 5). Representative images of wound confluency are provided in Appendix A.

### 3.4. Cell Signalling Effects

The abundance of key proteins involved in PI3K− and cell cycle−mediated cell signalling were determined following treatment with PIK-75 and/or dinaciclib as single agents and in combination (Figure 6).

Total AKT levels remained consistent across treatment types, whilst phosphorylated−AKT (pAKT) decreased in response to PIK-75 in both cell lines (Figure 6). Dinaciclib appears to have increased pAKT relative to the control, and consequently, it dampened the effects of PIK-75 in the combination.

P53 protein expression was not apparently altered for any condition with UW-CSCC1, whilst a reduction was observed in response to PIK-75 for UW-CSCC2. However, by 3 h, any impact was restored to normal.

The protein expression of CDK2 and CDK6 remained unaffected by any treatment, and the cyclin B1 blots did not provide a confident determination. Notably, cyclin D1 expression increased relative to the control in both cell lines upon exposure to PIK-75, whilst it decreased in response to dinaciclib at both time points. P21 expression was not evident in either cell line. P27 KIP1 was largely absent in UW-CSCC2, and dinaciclib exhibited some impact on P27 KIP1 in UW-CSCC1, although conclusive findings could not be drawn. Densitometry analysis of the bands observed in Figure 6 are in Appendix A.

### 3.5. Three−Dimensional Cell Viability Assays and Drug Synergy Studies

To provide a more physiologically relevant context, drug treatments and combinations were assessed against 3D spheroid models of the UW−CSCC cell lines, using an increase in green fluorescence intensity as a measure of cell death.

For UW-CSCC1, dinaciclib produced a greater impact than PIK-75, whilst the PIK:DIN mixture displayed a synergistic effect, providing the greatest degree of fluorescence (Figure 7a). Despite an underwhelming drug combination effect in 2D, BGT226 alone was interrogated in a 3D context finding a fluorescence intensity akin to the other PI3Ki, PIK-75. However, when BGT226 was mixed with dinaciclib, a substantial increase in fluorescence intensity was observed.

For UW-CSCC2, dinaciclib alone produced no effect relative to the control for any time point (Figure 7b). Further, whilst PIK-75 alone produced a time−dependent increase in fluorescence intensity, the combination with dinaciclib produced no significant difference, unlike in UW-CSCC1. BGT226 alone produced approximately double the fluorescence intensity of PIK-75 treated UW-CSCC2; however, the combination with dinaciclib also provided no discernible benefit.

A 3D drug synergy assay was subsequently completed to examine synergistic concentrations more closely (Appendix A). We found that dose optimisation may be required to elicit a greater synergistic effect for PIK:DIN with UW-CSCC1. PIK:DIN failed to demonstrate synergy with UW-CSCC2 spheroids at the concentrations tested (Appendix A), aligning with observations in Figure 7b. In contrast to the 2D drug synergy studies, BGT/DIN combinations produced strong synergy profiles for both cell lines (Appendix A). Together these results suggest an inversion of observations between a 2D and 3D context.

## 4. Discussion

In this study, our aim was to evaluate the potential therapeutic benefits of combining PI3K inhibitors with a cyclin−dependent kinase inhibitor in the context of metastatic cSCC using relevant cell line models of the disease. Metastatic cSCC poses a significant clinical challenge due to its aggressive nature and limited treatment options. Given the emerging evidence implicating the PI3K signalling pathway in cSCC progression and its proposed combination with CDKs, we hypothesised that dual targeting of these pathways may synergistically inhibit cell proliferation and impede metastatic potential. We have observed that at specific concentrations, the PI3K inhibitors, chiefly PIK-75, work mostly additively or synergistically with dinaciclib in impacting cell viability, apoptosis, and cell motility, depending upon the cell line interrogated. However, cell cycle distribution and key molecular signalling pathways involved in cSCC progression remained largely unaffected by the drug combination in comparison to the drug as a single treatment at the chosen concentrations and time points. A summary of these observations can be found in Table 3.

A simplified proposed mechanism to explain these effects is provided in Figure 8.

### 4.1. UW-CSCC1 and UW-CSCC2 Present Distinct Molecular Profiles for Cell Cycle− and PI3K−Related Pathways

Key genes pertaining to cell cycle and PI3K signalling were analysed between the two cell lines, finding distinct profiles. Of note, despite recurrent alterations observed in cSCC elsewhere [47,48], both cell lines displayed wild−type *PIK3CA* status. Whilst constitutive activation of PI3K can arise from multiple mechanisms, including mutations in *PIK3CA*, loss of negative regulators (e.g., *PTEN* or *PIK3R1)*, and RAS activation [49,50,51], it is unlikely that these are the driving force in our cSCC cell lines. Despite this, we observed the constitutive expression of PI3K downstream effectors. We propose the activation of receptor tyrosine kinases and increased mitogenic signalling upstream of PI3K in general as the reasoning for this along with potential miRNA and long non−coding RNA effects.

Whilst a *TGFBR1* stop-gained mutation was found in UW-CSCC1, the mutation occurs at position 301 of 500, enabling 60% of the protein to be transcribed. Coupled with the greater CNV than that of UW-CSCC2, this may explain the null effect on gene expression. While stop-gained mutations generally correlate with a decrease in RNA expression relative to the wild−type counterpart, our Magpix analyses (Appendix A) highlight the incongruence between mRNA and translated protein data.

UW-CSCC1 possessed an in−frame insertion mutation in *RB1*, whilst UW-CSCC2 did not. This mutation may disrupt the function of the RB1 protein for UW-CSCC1, and loss of function has been associated with resistance to CDK4/6 inhibitors [52]. However, we noted no significant difference in cell viability response to the CDKi dinaciclib between cell lines, as dinaciclib targets CDK1/2/5/9, not CDK4/6. We hypothesise that UW-CSCC1 may be less responsive to CDK4/6 inhibitors than UW-CSCC2 because of *RB1* mutation, thus supporting the use of dinaciclib in this context. The main targets of dinaciclib responsible for the cell cycle modulating effect, CDK1 and CDK2, are present in three copies and at similar levels in both cell lines, aligning with the similar sensitivities observed to dinaciclib. Whilst a broader pan−CDK inhibitor such as flavopiridol would safeguard the molecular variability of a patient’s tumours, they carry their own drawbacks [29] and it may be safer to utilise selective inhibitors specific to the individual.

The tumorigenic impact of each of the somatic variants reported and their influence on response to PI3K or CDK inhibition cannot be readily ascertained without a more comprehensive transcriptomic analysis and acknowledgement of the influence of miRNA, post−translational modification, and epigenetics [53]. Such functional genomic analyses go beyond the scope of this investigation, yet it is reasonable to presume the presence of these recurrent variants supports their candidacy as drug targets for mcSCC. Together, these findings highlight the genetic heterogeneity of mcSCC and suggest potential targets based on individual profiles.

### 4.2. PI3K and Cell Cycle Inhibitors Potently Reduced cSCC Cell Viability In Vitro

Both cell lines possessed a low nanomolar sensitivity towards PIK-75, BGT226, and dinaciclib as monotherapies, whilst the keratinocyte cell line HaCaT displayed no sensitivity at the concentrations tested (0–1 µM). Of interest, UW-CSCC2 displayed greater sensitivity to the selective PIK-75 than UW-CSCC1, despite a similar sensitivity to the pan−PI3K inhibitor BGT226. While both cell lines bear no mutation in *PIK3CA,* different copy numbers of the gene in UW-CSCC1 (4.60) and UWCSCC2 (3.39) as well as increased overall expression levels of p110α in UW-CSCC1 (Appendix A) provide an explanation for the decreased sensitivity. Regardless of subsequent synergy analyses, these data provide further evidence on the efficacy of these drugs in targeting mcSCC.

### 4.3. Synergistic Effect of PI3K and CDK Dual Inhibition

Two−dimensional checkerboard assays and Fa−CI analysis revealed a synergistic interaction between PIK-75 and dinaciclib in both cell lines, with specific concentration ratios maximising this effect. Notably, UW-CSCC2 demonstrated a stronger synergistic response compared to UW-CSCC1, potentially due to its distinct genetic alterations and signalling pathways. In stark contrast, BGT226 and dinaciclib combinations in 2D were largely antagonistic. The broader inhibition of PI3K and mTOR by BGT226 may simultaneously disrupt multiple pro−survival pathways, preventing synergistic interactions with dinaciclib.

It has been well characterised that anti−cancer agents have a different sensitivity in 2D and 3D culture systems [54]. On this basis, we assessed drug treatments against the spheroids using a four−fold increase in concentration and found the relationships observed in 2D somewhat inverted in 3D. Cell–cell interactions in a 3D context may be affecting signalling responses and therefore altering cell survival. The penetrability of the drugs into spheroids should also be determined in future investigations to resolve their candidacy for pre−clinical study.

### 4.4. Mechanism of Action and Downstream Effects

While the PIK:DIN combination failed to significantly increase apoptosis compared to single agents, both cell lines displayed elevated pAKT levels upon dinaciclib treatment. This suggests that dinaciclib might counteract the PIK-75−mediated AKT inhibition observed, potentially explaining the lack of a synergistic effect on apoptosis. This phenomenon has been seen with CDK4/6 inhibitors inducing PI3K/AKT pathway activation, hence the drawback of using CDK inhibitors as a monotherapy [55,56]. Further, the high proportion of cells in either late apoptosis or necrosis suggests our time point of 24 h may be too delayed to witness a synergistic effect. The proportion of dead cells gated in the combination samples was substantial, indicating cells were disproportionately completing apoptosis prior to analysis. A shorter time point of 6 h was investigated, but the results were inconclusive.

Cell cycle analysis in both cell lines revealed increased proportions of cells in the S+G2 phase in response to both PIK-75 and dinaciclib treatment. Despite the direct effect of dinaciclib on cell cycle regulators, it appeared that PIK-75 was the dominating driver of cell cycle stalling. This response is consistent with a proposed mechanism of S+G2 arrest via nucleoside depletion in response to PI3K inhibition [57,58].

The combination of PIK-75 and dinaciclib resulted in a significant reduction in cell motility for both cell lines, although the effect appeared dominated by dinaciclib. One explanation is that dinaciclib targets CDK5, a known regulator of cancer cell motility [59,60], which in our cell lines override PIK-75 effects, even though PI3Ks play key roles in regulating cell motility [61,62].

To gain a better insight into the mechanism of the combination therapy and its impact upon pathway signalling, relevant protein expression was examined via Western blot. As stated above and previously shown by us [39], PIK-75 is effective in reducing pAKT protein expression. Regarding cyclin B1, no conclusive results could be determined from the blots, indicating that the treatments did not have a significant impact on its expression. Further investigation using alternative techniques or additional time points may be warranted to better understand the regulation of cyclin B1 in response to PIK-75 and dinaciclib treatments. Despite dinaciclib also targeting CDK2, no change in protein expression was observed for any treatment type. Given the cyclical nature of CDKs, a longer timeframe may be required to observe an impact. Notably, CDK2 demonstrated a second band with UW-CSCC1 that we have been unable to find any evidence of elsewhere, perhaps a result of post−translational modifications.

Interestingly, cyclin D1 expression increased relative to the control across both cell lines specifically in response to PIK-75 treatment. Generally, PI3K inhibition will result in a reduction in cyclin D1 [63,64], although some form of compensatory upregulation of cyclin D1 as a mechanism to promote cell cycle progression has occurred.

The mutations in *TP53* genes of both cell lines are reportedly pathogenic, as indicated by the high CADD scores, and are associated with the loss of function of p53 [65]. The reduced stability induced by the resulting amino acid changes leads to the accumulation of misfolded p53. This is consistent with the constitutive expression in the Western blots and is possibly masking changes in p53 levels in response to drug treatments. For example, the inhibition of PI3K can upregulate p53 mRNA and protein expression levels [66]; however, we have reported a decrease in this instance. GAPDH loading controls for this blot confirmed equal protein loading across lanes, and therefore, this result is anomalous.

The faint bands observed for P21 and P27 KIP1 suggest low expression levels of these proteins in the cSCC cell lines. These proteins are key regulators of cell cycle progression, acting themselves as CDK inhibitors. The low expression levels may indicate a lack of active cell cycle arrest mechanisms in these cell lines, or potential post−translational modifications affecting the stability. As cell cycle regulators, these proteins are primarily located in the nucleus [67], and therefore, subcellular fractionation may be required to better isolate proteins.

### 4.5. Limitations and Future Directions

The number of cell lines included and the lack of validation in more clinically relevant models such as murine models or ex vivo cultures limit the translatability of this study. In order to pinpoint mutations that convey susceptibility to PI3Ki, CDKi, and their combination, studies involving functional genomics, such as CRISPR knockout screens, could be employed. These allow for genome−wide screening of mutations that convey chemoresistance as well as mutations that increase susceptibility to therapeutic regimen [68,69]. Validating the findings from those studies can be challenging as the use of CDKi, PI3Ki, and their combination is not currently approved for use in cSCC. However, ex vivo cultures of mcSCC could provide an elegant solution to this dilemma. Concretely, the drug treatment of ex vivo cultures derived from surgical specimens could be correlated with genetic sequencing and validate hits identified in the CRISPR knockout screen [70,71]. Ex vivo cultures have been successfully investigated in other cancers including haematological malignancies, breast cancer, and HNSCC for their use in drug response profiling [72,73,74]. There have been some efforts to translate this methodology to cSCC. However, such efforts were confined to primary cSCC, which can be easily treated via surgical excision [75,76], and require expansion to mcSCC samples to increase their relevance in this challenging to treat disease.

## 5. Conclusions

This study offers valuable insights into the complex interplay between PI3K and CDK inhibitors in cSCC, contributing to a deeper understanding of potential synergism. Further research targeting these pathways holds promise for developing more effective combination therapies, particularly in those groups for whom checkpoint inhibitors are contraindicated. While some responses were underwhelming, overall, our study presents promising evidence for the synergistic potential of select PI3Ki with the CDKi dinaciclib. Given this, we maintain that a PI3Ki–CDKi combination warrants continued investigation, although the challenge lies in pairing the isoform specificity of the drug to the mutational profile of the cancer and capturing the optimal drug combination dosage. Importantly, we observed differential responses based on individual cell line profiles, highlighting the need for personalised medicine approaches in tailoring therapeutic strategies for patients with mcSCC.

## Figures and Tables

**Figure 1 cancers-16-00370-f001:**
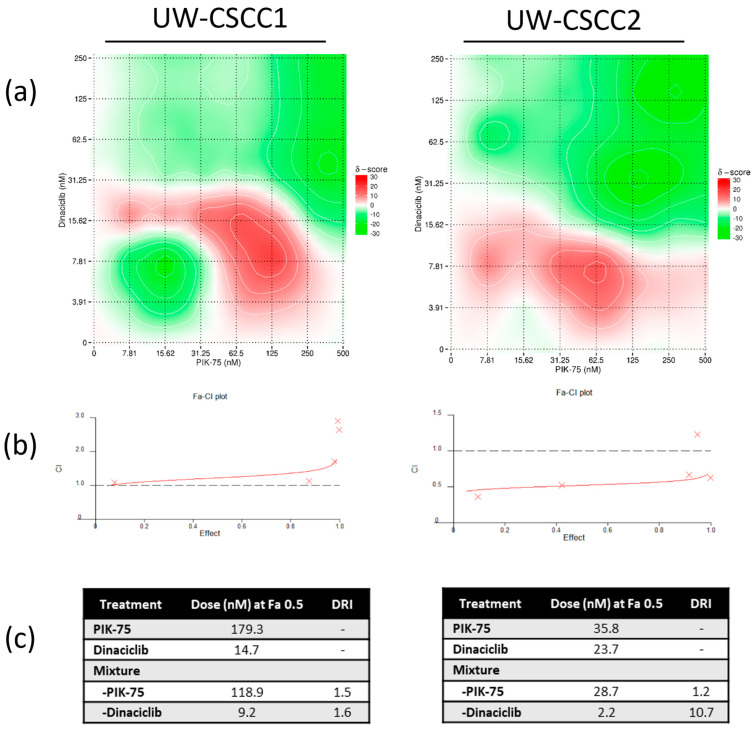
Synergistic effect profile of combination PIK-75 (selective PI3Ki) and dinaciclib (CDKi) on UW-CSCC1 and UW-CSCC2 cell lines in 2D culture conditions. (**a**) Synergistic score matrix of drugs after 72 h, implementing a BLISS analysis model (*n* = 3). BLISS synergy scores (δ): <−10 signifies antagonism, >−10 but <10 signifies an additive interaction, and >10 signifies synergy. (**b**) Combination index (Fa−CI) plot using the Chou–Talalay median−effect equation. A score >1 indicates antagonism, a score <1 indicates synergism, and a score = 1 indicates an additive effect. As effect increases, estimates of synergy/antagonism become weaker. (**c**) Dose–effect table indicating dose required to elicit a 50% fraction affected (Fa) response. Dose−reduction index (DRI) at Fa 0.5 for the drug combinations is shown.

**Figure 2 cancers-16-00370-f002:**
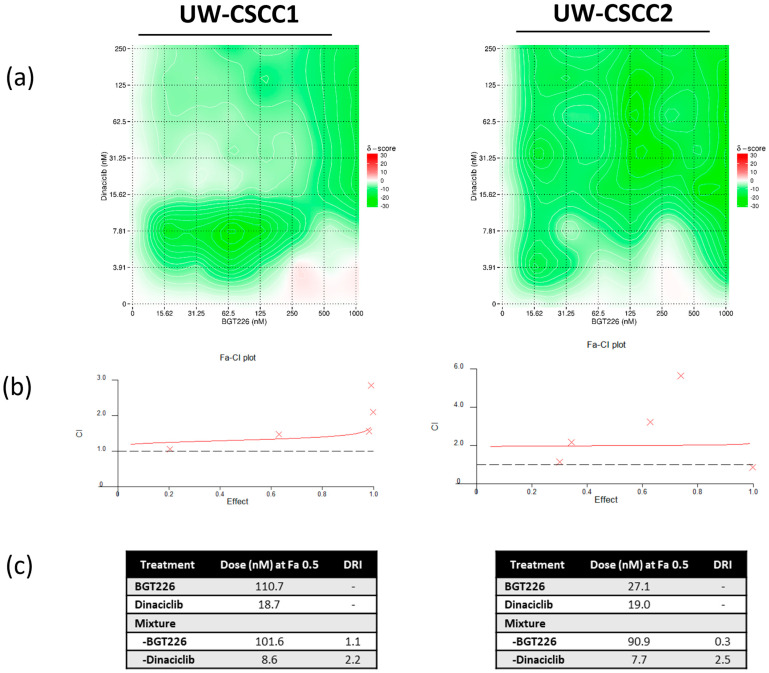
Synergistic effect profile of combination BGT226 (pan−PI3Ki) and dinaciclib (CDKi) on UW-CSCC1 and UW-CSCC2 cell lines. (**a**) Synergistic score matrix of drugs after 72 h, implementing a BLISS analysis model (*n* = 2). BLISS synergy scores (δ): <−10 signifies antagonism, >−10 but <10 signifies an additive interaction, and >10 signifies synergy. (**b**) Combination index (Fa−CI) plot using the Chou–Talalay median−effect equation. A score >1 indicates antagonism, a score <1 indicates synergism, and a score = 1 indicates an additive effect. As effect increases, estimates of synergy/antagonism become weaker. (**c**) Dose–effect table indicating dose required to elicit a 50% fraction affected (Fa) response. Dose−reduction index (DRI) at Fa 0.5 for the drug combinations is shown.

**Figure 3 cancers-16-00370-f003:**
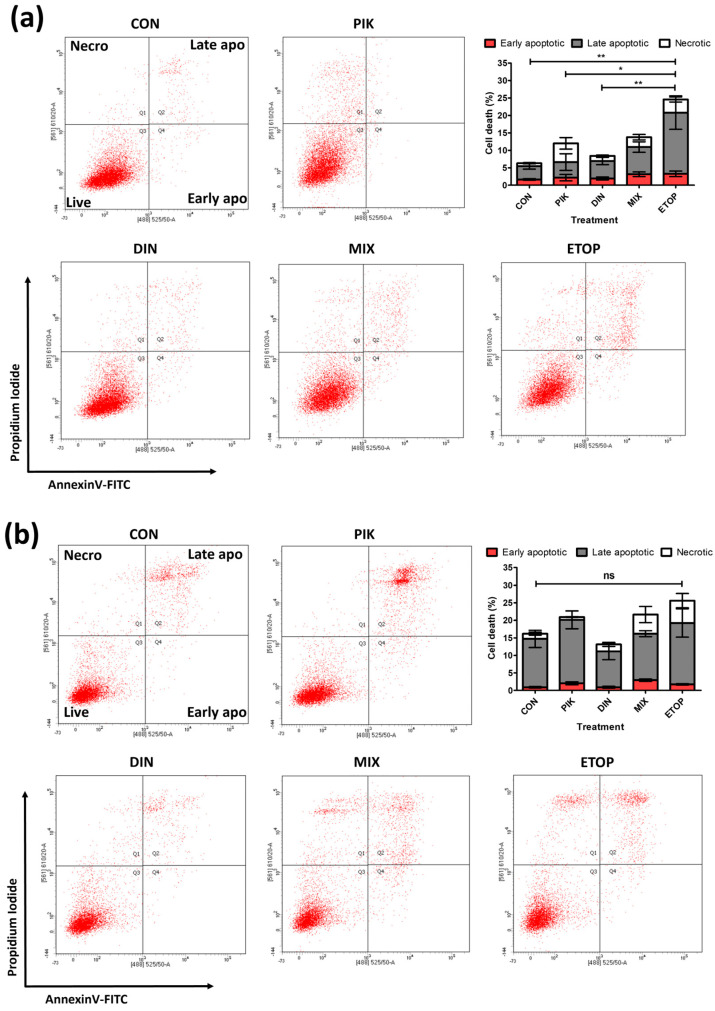
Early− and late−stage apoptosis rates across treatment for UW-CSCC1 (**a**) and UW-CSCC2 (**b**). Cells were treated with vehicle (DMSO), PIK-75 (both cell lines: 62.5 nM), dinaciclib (UW-CSCC1: 16 nM; UW-CSCC2: 8 nM), or a combination. After 24 h of treatment, apoptosis in both cell lines was determined by flow cytometry after staining with annexin V−FITC/PI. CON, control; PIK, PIK-75; DIN, dinaciclib; MIX, mixture; ETO, etoposide. Standard error of the mean is shown, *n* = 4 (UW-CSCC1) and *n* = 5 (UW-CSCC2). A one−way ANOVA was applied along with Tukey’s multiple comparison post−test. Significance denoted by asterisks: * = *p* < 0.05, ** = *p* < 0.01, ns = not significant.

**Figure 4 cancers-16-00370-f004:**
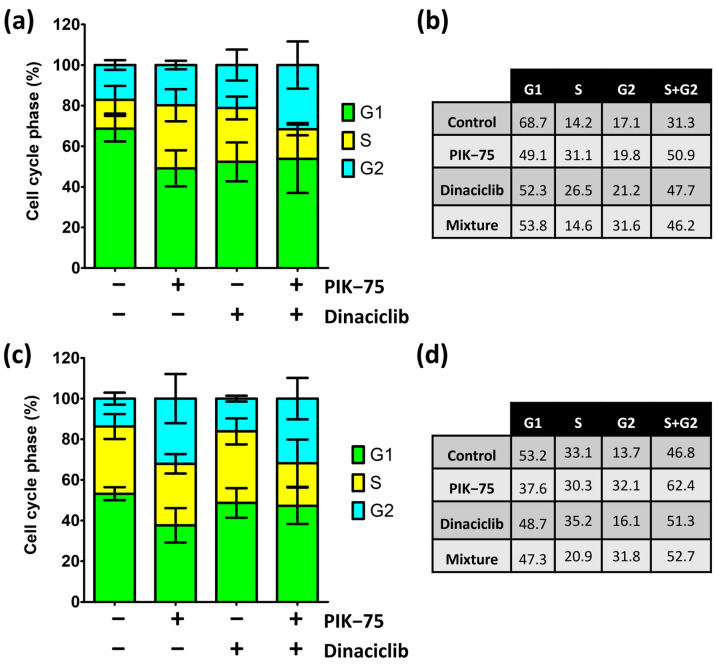
Cell cycle phase distribution of UW-CSCC1 and UW-CSCC2 ± treatment. Results have been scaled to 100% not including events detected <G0 and >G2. (**a**,**c**): cell cycle phase distribution of UW-CSCC1 and UW-CSCC2, respectively, in response to 48 h with different treatments: vehicle control (DMSO), PIK-75 (both cell lines: 62.5 nM), dinaciclib (UW-CSCC1: 16 nM; UW-CSCC2: 8 nM), or in combination. The percentage of cells in a specific cell cycle phase (G1, S, or G2), are represented as stacked bars (*n* ≥ 3). Error bars represent the standard error of the mean. (**b**,**d**): UW-CSCC1 and UW-CSCC2 cell cycle distribution as a normalised percent.

**Figure 5 cancers-16-00370-f005:**
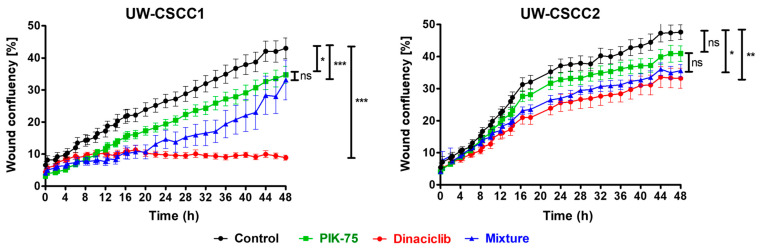
Random migration of UW-CSCC1 and UW-CSCC2 in response to PI3Ki and CDKi combination or monotherapy. Wound confluency shown over 48 h as assessed by scratch wound assay in the presence of PIK-75 (both cell lines: 62.5 nM) or dinaciclib (UW-CSCC1: 16 nM; UW-CSCC2: 8 nM), or their combination, as well as an untreated control. Standard error of the mean is shown, *n* = 10. A one−way ANOVA was applied along with Tukey’s multiple comparison post−test. Significance denoted by asterisks: * = *p* < 0.05, ** = *p* < 0.01, *** = *p* < 0.001, ns = not significant.

**Figure 6 cancers-16-00370-f006:**
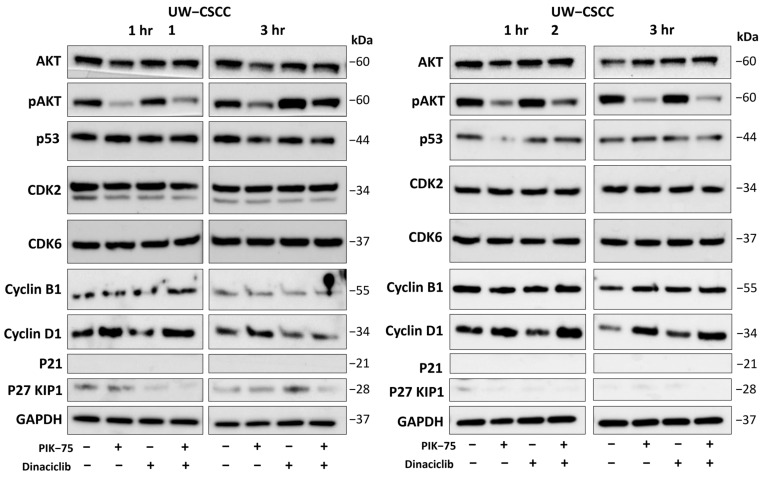
Expression of key cell cycle regulators and components of the PI3K signalling pathway in response to treatment against UW-CSCC1 (**left**) and UW-CSCC2 (**right**). PIK-75 (both cell lines: 62.5 nM); dinaciclib (UW-CSCC1: 16 nM; UW-CSCC2: 8 nM). The uncropped bolts are shown in Appendix A.

**Figure 7 cancers-16-00370-f007:**
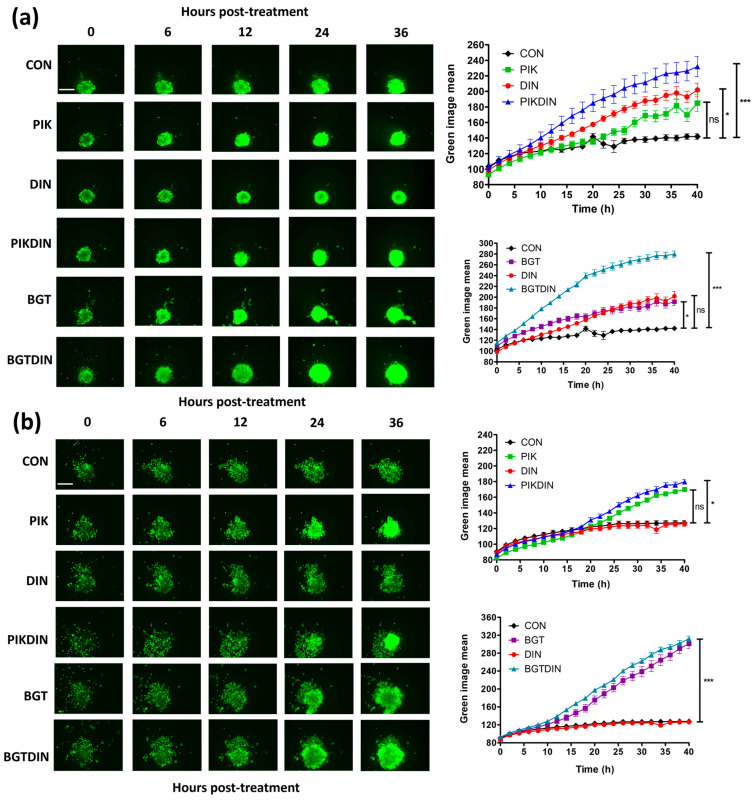
Cytotoxicity in UW-CSCC1 (**a**) and UW-CSCC2 (**b**) spheroids treated with PI3Ki (PIK-75 or BGT226), dinaciclib, or a combination. Increased Cytotox Green staining indicates increased cell death in the combination treatment compared to DMSO control or single−agent treatment. Representative images at select time points are provided. Standard error of the mean is shown, *n* = 10. CON, control (DMSO); PIK, PIK-75 (250 nM); DIN, dinaciclib (UW-CSCC1: 64 nM; UW-CSCC2: 32 nM); PIKDIN, PIK-75 and dinaciclib mixture; BGT, BGT226 (1000 nM); BGTDIN, BGT226 and dinaciclib mixture. A one−way ANOVA was applied along with Tukey’s multiple comparison post−test. Significance denoted by asterisks: * = *p* < 0.05, *** = *p* < 0.001, ns = not significant. The scale bar (top left image in the panel) represents 400 μm).

**Figure 8 cancers-16-00370-f008:**
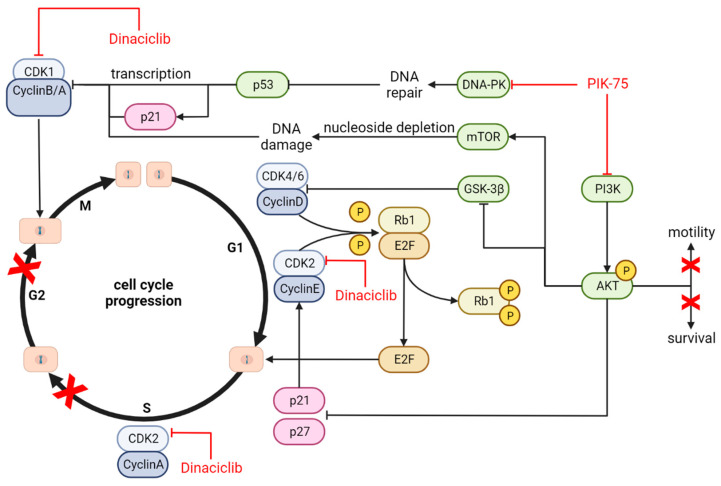
Key signalling pathways targeted by a combination of PI3Ki (PIK-75) and CDKi (dinaciclib). Dinaciclib inhibits the progression through the cell cycle by supressing the signalling of CDK2 (G1/S and S/G2 transition) and CDK1 (G2/M transition) and stalling at the respective checkpoints. PIK-75, a dual inhibitor of DNA−PK and PI3K, also can promote cell cycle arrest in G2− or G1−phase via the induction of DNA damage or through interference with CDK4/6 (AKT−GSK−3β−CDK4/6 axis), respectively. Independently of the cell cycle modulating effect of PIK-75, PIK-75 can inhibit cell motility and reduce cell survival by decreasing phospho−AKT levels. Red crosses indicate cellular perturbation elicited by the drug treatment.

**Table 1 cancers-16-00370-t001:** Coding mutations in UW-CSCC1 and UW-CSCC2 for commonly altered cell cycle and PI3K/AKT/mTOR genes. Combined annotation−dependent depletion (CADD) scores which serve as a measure of the predicted deleteriousness of the mutation. Scores < 20 indicate minimal impact. Transcript per million (TPM), a measure of the gene’s expression level, is also indicated along with copy number variation (CNV).

	Gene	UW-CSCC1				UW-CSCC2			
		Mutation Type	Impact (CADD)	TPM	CNV	Mutation Type	Impact (CADD)	TPM	CNV
Cell cycle	*CCNB3*			1.4	1.98	missense	1.945	6.3	1.89
	*CDK12*	synonymous	10.79	51	3.98			60	3.80
		synonymous	7.65						
	*CDKN2A*			451	3.81	stop-gained	n.a.	124	2.83
	*NOTCH1*			14	4.06	stop-gained	48	7	5.83
						synonymous	6.043		
	*NOTCH2*			67	3.03	stop gained	41	11	1.37
	*PAK4*	missense	25.6	99	4.92			109	3.03
	*PAK5*	missense	31	0.05	4.02			0.2	6.73
	*RB1*	in−frame insertion	n.a.	28	2.98			37	2.46
	*TP53*	missense	29.5	109	3.15	stop-gained	36	153	3.38
		missense	31			missense	28.4		
	*WEE1* *			46	3.15	stop-gained	n.a.	61	3.29
PI3K/AKT/mTOR	*AKT3*	missense	28.2	62	3.01	synonymous	12.69	22	2.92
	*EGFR*	missense	26	20	2.31			196	3.47
	*ERBB2*	synonymous	10.74	80	3.98			70	3.80
	*ERBB3*	missense	25.6	5.7	3.01	missense	23.4	21	2.94
		missense	23.5						
	*HRAS*			20	3.15	missense	n.a.	57	3.29
	*PIK3C2A*			27	3.15	missense	17.23	36	3.29
	*PIK3C2B*	missense	26.3	5.3	3.01			20	2.92
	*PIK3CG*	missense	24.1	2.4	3.04	missense	20.7	1.4	2.98
	*PIK3R5*	missense	22.7	0.11	3.15			0.01	3.38
	*PIK3R6*	frameshift	n.a.	0.04	3.15			0.04	3.38
		complex substitution	n.a.						
		missense	16.13						
		missense	n.a.						
	*SMAD3*			103	2.09	complex substitution	n.a.	233	3.99
	*TGFBR1*	stop gained	38	73	4.92			29	3.03

n.a. = not available; * non−recurrent functionally significant alteration.

**Table 2 cancers-16-00370-t002:** Cytotoxic response of UW-CSCC1 and UW-CSCC2 to PIK-75, BGT226, and dinaciclib in 2D culture. Inhibitory concentration 50 (IC_50_), the value at which cell viability is 50%, is shown along with the mean ± SD of *n* ≥ 3 experiments each performed in triplicate. Asterisks indicate statistically significant differences between cell lines (*p* < 0.001). NA = not assessed.

Cell Line	IC_50_ (nM)		
	PIK-75	BGT226	Dinaciclib
UW-CSCC1	220 ± 32 *	223 ± 43	19 ± 2.8
UW-CSCC2	56 ± 3 *	195 ± 42	20 ± 2.4
HaCaT	No effect	NA	No effect

**Table 3 cancers-16-00370-t003:** Summary of cell effects in response to PI3K−CDK drug combinations.

Variable	UW-CSCC1		UW-CSCC2	
	PIK:DIN	BGT/DIN	PIK:DIN	BGT/DIN
2D synergy assay	Strongly additive	Weakly additive	Synergistic	Antagonistic
3D synergy assays	Additive	Synergistic	Weakly additive	Synergistic
Apoptosis	Weakly additive, PIK-75 dominates	Not Assessed	Weakly additive, PIK-75 dominates	Not Assessed
Cell cycle phase	No synergy, PIK-75 effect dominates	Not Assessed	No synergy, PIK-75 effect dominates	Not Assessed
Motility	Weakly additive, dinaciclib effect strongly dominates	Not Assessed	Weakly additive, dinaciclib effect dominates	Not Assessed
Cell signalling	No synergy, some effects dominated by either drug	Not Assessed	No synergy, some effects dominated by either drug	Not Assessed

## Data Availability

Data are available from the corresponding authors upon reasonable request.

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
