# Peer review of "PIK Your Poison: The Effects of Combining PI3K and CDK Inhibitors against Metastatic Cutaneous Squamous Cell Carcinoma In Vitro"

_cancers, 2024, doi:10.3390/cancers16020370_

Round 1

Reviewer 1 Report

Comments and Suggestions for Authors

This is a highly quality presentation, which, however, would benefit from some revisions.

The manuscript for the most part is well written and data were properly collected using sound methodologies. I do not have a major critique here. The critique is more on interpretation, discussion and clarification.

The authors used SCC lines established from metastases by not derived from localized to the skin tumors, why?

Accordingly, discussion could be enhanced by indicating that the implications are for metastatic disease, but more work is still to be done to have better therapeutic approach for localized to the skin disease, aside of the surgery.

Also, mentioning that advanced (metastatic) tumors can regulate their environment and body homeostasis as recently discussed (How cancer hijacks the body’s homeostasis through the neuroendocrine system, Trends Neurosci 46: 263-275, 2023. https://doi.org/10.1016/j.tins.2023.01.003.). Topic relevant to the subject of the paper.

Addition of microscopic bars to some images would be appreciated.

Section on limitations of the study would enhance the paper.

Author Response

We thank the reviewer for their comments.

The reason for using cSCC cell lines derived from metastases rather than from localised (primary) tumors was because they best represent the mcSCC patient demographic for which novel treatments are needed. We did note in our Introduction that mcSCC is associated with worse prognosis and higher risk of disease specific death than localised disease due to limited effective therapeutic options. Furthermore, mcSCC is more frequent in immunosuppressed patients in which immune checkpoint inhibitors are counter indicated. Consequently, we sought to utilize our mcSCC cell lines as models that best represents the mcSCC patient demographic to evaluate the benefit of combining the PI3Ki, PIK-75, with the CDKi, dinaciclib.

We have adapted the Introduction and Discussion to better reflect this rationale.

While interesting, we respectfully disagree that the suggested research article is particularly relevant to our study. While we employ a metastatic model, the focus of the study is in vitro and on cellular molecular biology and the review is focussed on a physiological level.

We have added a mention of the scale bars in the respective figure legend, as requested.

We have added a section on limitations to the discussion.

Reviewer 2 Report

Comments and Suggestions for Authors

-       The authors investigated the potential of PI3K inhibitors (PI3Ki) and cell-cycle inhibitors (CDKi) as single-agents and in combination against patient-derived mcSCC cell lines. Whilst PI3Ki and CDKi as single agents potently induced cancer cell death, PI3Ki synergistically enhanced the potential of dinaciclib to induce cell death in one mcSCC cell line, but not another.

-       This study is very interesting and novel and fills a gap in the current literature suggesting that personalized medicine approaches targeting PI3K and CDK 20 pathways in combination may yield some benefit.

-       Methods are appropriate and adequately described.

-       Results are relevant.

-       Figure and tables are very descriptive.

-       Nevertheless, the paper lacks a discussion section with the authors’ critical insight. Please add.

Author Response

We thank the reviewer for their comments. All changes and additions have been implemented as requested.

Reviewer 3 Report

Comments and Suggestions for Authors

The work demonstrates the synergistic activity of combining PI3K- and CDK-in- 2 hibitors against metastatic cSCC in vitro. The study is well written and well-characterized, and I would like to thank the authors for this great work. Only few comments from my side:

1- Please state the source of all materials including the tissue culture media components in the Materials section

2-  Title 3.2., please correct the word synergy

3- Please unify the abbreviation of hours to (h) throughout the manuscript

Author Response

(The authors gave the same response as above.)
